# Carvacrol Inhibits Quorum Sensing in Opportunistic Bacterium *Aeromonas hydrophila*

**DOI:** 10.3390/microorganisms11082027

**Published:** 2023-08-07

**Authors:** Liushen Lu, Junwei Wang, Ting Qin, Kai Chen, Jun Xie, Bingwen Xi

**Affiliations:** 1Key Laboratory of Aquatic Animal Nutrition and Health, Freshwater Fisheries Research Center, Chinese Academy of Fishery Science, Wuxi 214081, China; luliushen663@163.com (L.L.); qint@ffrc.cn (T.Q.); chenk@ffrc.cn (K.C.); xiej@ffrc.cn (J.X.); 2Wuxi Fisheries College, Nanjing Agricultural University, Wuxi 214128, China; wangjwnjau@163.com

**Keywords:** *A. hydrophila*, carvacrol, N-acyl-homoserine lactones, pathogenicity, quorum sensing

## Abstract

Bacterial quorum sensing (QS) plays a crucial role in chemical communication between bacteria involving autoinducers and receptors and controls the production of virulence factors in bacteria. Therefore, reducing the concentration of signaling molecules in QS is an effective strategy for mitigating the virulence of pathogenic bacteria. In this study, we demonstrated that carvacrol at 15.625 μg/mL (1/4 MIC), a natural compound found in plants, exhibits potent inhibitory activity against QS in *Chromobacterium violaceum*, as evidenced by a significant reduction (62.46%) in violacein production. Based on its impressive performance, carvacrol was employed as a natural QS inhibitor to suppress the pathogenicity of *Aeromonas hydrophila* NJ-35. This study revealed a significant reduction (36.01%) in the concentration of N-acyl-homoserine lactones (AHLs), a QS signal molecular secreted by *A. hydrophila* NJ-35, after 1/4 MIC carvacrol treatment. Moreover, carvacrol was found to down-regulate the expression of *ahyR/I*, two key genes in the QS system, which further inhibited the QS system of *A. hydrophila* NJ-35. Finally, based on the above results and molecular docking, we proposed that carvacrol alleviate the pathogenicity of *A. hydrophila* NJ-35 through QS inhibition. These results suggest that carvacrol could serve as a potential strategy for reducing the virulence of pathogenic bacteria and minimizing the reliance on antibiotics in aquaculture.

## 1. Introduction

*Aeromonas hydrophila*, a significant pathogen of fish, amphibians, reptiles, and mammals, widely distributed in freshwater environments, is commonly associated with fish diseases such as furunculosis, ulceration, and hemorrhagic septicemia [1]. These infections are usually treated by antibiotics in aquaculture. However, the excessive use and even abuse of antibiotics not only pollute the environment but also induce antibiotic resistance [2]. These issues have raised awareness that the use of antibiotics should be more cautious [3]. In addition, *A. hydrophila* is resistant to various antibiotics [4]. Therefore, alternative methods for the prevention or treatment of these infections are necessary for the sustainable development of the aquaculture industry.

One promising approach is disrupting bacterial quorum sensing (QS) because QS is a chemical-based language developed by bacteria, which can coordinate the expression of virulence factors in pathogenic bacteria in the host organism [5]. QS is a bacterial signaling mechanism that utilizes cell–cell communication and population level behavior in response to cell density [6,7,8]. It is activated through a sequential process by binding to a specific receptor protein with an extracellular signal called an autoinducer [9]. When the concentration of the autoinducer accumulates in the environment to a critical level, it can bind with the QS receptor, leading to the massive expression of virulence genes and the production of virulence factors, which enable the bacteria to infect their hosts [10]. This system has been found to control the production of various extracellular virulence factors and biofilm formation in bacterial pathogens such as *A. hydrophila*; therefore, blocking the bacterial QS mechanism can disrupt QS-regulated pathogenic phenotypes, which can ultimately reduce pathogenicity.

The N-acyl homoserine lactone (AHL) system was initially found in marine bacteria *Vibrio fischeri* to control the production of luminescence [11]. AHL is a signaling molecule that plays a crucial role in mediating the autoinducer type 1 QS system of Gram-negative bacteria [12]. The AHL molecules produced by each type of bacteria are different, mainly due to variations in the length of the fatty acid chain. Therefore, each type of bacteria can only recognize the AHL molecule that they produce, hence the AHL system exhibits species-specificity [13]. Previous research has indicated that *A. hydrophila* secretes two types of AHLs, namely N-3-butanoyl-DL-homoserine lactone (C4-HSL) and N-3-hexanoyl-DL-homoserine lactone (C6-HSL), with C4-HSL being the more prominent [14]. Regulating the concentration of these AHL signaling molecules has proven to be an effective strategy for inhibiting the QS system [15,16]. A QS inhibitor can interfere with AHL-dependent QS systems, leading to a decrease in the production of virulence factors [17,18]. Therefore, compounds, including synthetic drugs and natural phytochemicals, have attracted significant interest for their potential to interfere with AHL-mediated QS systems, with the latter being particularly noteworthy due to their high efficacy and low toxicity. For example, small molecular compounds (such as polyphenols and flavonoids) extracted from 14 medicinal plants and spices exhibited anti-QS effects [19]. In addition, previous research has demonstrated that carvacrol, the principal component of oregano essential oil extracted from the plant *Origanum vulgare*, can effectively inhibit the growth and virulence of *A. hydrophila* at sub-inhibitory concentrations [20,21]. Despite its widespread use in the food industry, the effectiveness of carvacrol as an antimicrobial agent has only been partially explored. For example, carvacrol can significantly reduce the production of pyocyanin in *Pseudomonas aeruginosa* by 60% and exhibit promising anti-QS potential [22]. Therefore, carvacrol could potentially serve as a natural QS inhibitor to mitigate the toxicity of pathogenic bacteria.

In this study, we demonstrated that carvacrol acted as a natural QS inhibitor using *Chromobacterium violaceum* CV026, which was a biosensor bacterium strain and produced violacein in the presence of an exogenously supplied short chain AHL, and further extended its application to reduce the toxicity of *A. hydrophila*. Moreover, to investigate the mechanism of carvacrol reducing the pathogenicity of *A. hydrophila*, we studied the concentration changes in AHL signal molecules in the *A. hydrophila* QS system as well as the expression of related genes. Additionally, we performed a simulation analysis to examine the binding mode between carvacrol and the receptor protein and calculated their binding free energy. This work provided the theoretical foundation for novel antibacterial therapy in aquaculture.

## 2. Materials and Methods

### 2.1. Bacterial Strain and Growth Condition

The clinical strain *A. hydrophila* NJ-35, isolated from diseased carp, was donated by Prof. Yong-Jie Liu (College of Veterinary Medicine, Nanjing Agricultural University, Nanjing, China). The quorum biosensor strain *C. violaceum* CV026, kept in the lab, was used in this study [23]. *A. hydrophila* NJ-35 and *C. violaceum* CV026 were maintained on Luria–Bertani (LB) broth at 28 °C.

Carvacrol (>99% HPLC purity; CAS no. 499-75-2) and N-3-butanoyl-DL-homoserine lactone (C4-HSL) were purchased from Aladdin (Shanghai, China). Dimethyl sulfoxide (DMSO) was bought from Sigma (China). LB medium was obtained from Shanghai yuanye bio-technology Co., Ltd. (Shanghai, China). Ethyl acetate was bought from Sinopharm Chemical Reagent (Beijing, China). Carvacrol was dissolved in DMSO to obtain a stock solution of 20.48 mg/mL and then diluted with LB or sterile distilled water.

### 2.2. Determination of Minimal Inhibitory Concentration (MIC) of Carvacrol

The MICs were determined using the broth microdilution method [24,25]. Briefly, *A. hydrophila* NJ-35 and *C. violaceum* CV026 were, respectively, inoculated into fresh LB broth to achieve a concentration of 1 × 10^6^ CFU/mL, and we then administered 90 µL strains to a 96-well plate, respectively. The following step took 10 µL different concentrations of carvacrol in the plate. The final concentrations of carvacrol were 0, 7.8125, 15.625, 31.25, 62.5, 125, 250, and 500 µg/mL. The plates were incubated at 28 °C, 180 rpm aeration, for 24 h. Negative and positive controls consisted of wells containing only LB and wells containing LB including bacteria, respectively. A concentration of carvacrol resulting in no visible growth was selected as the MIC.

### 2.3. Quantitative Assay of Violacein Production in C. violaceum CV026

This assay quantified the production of violacein as a proxy for QS activity. After the overnight incubation of *C. violaceum* CV026, the bacterial suspension (1 × 10^8^ CFU/mL, 1:100 *v*/*v*) was inoculated into fresh LB broth containing various concentrations of carvacrol (1/2 MIC = 31.25 μg/mL, 1/4 MIC = 15.625 μg/mL, 1/8 MIC = 7.8125 μg/mL, and 1/16 MIC = 3.90625 μg/mL) and C4-HSL (40 μg/mL) [26]. In addition, experimental groups of carvacrol were established without C4-HSL to evaluate the growth of *C. violaceum* CV026 (OD_600_). The negative control was added 1% DMSO. The cultures were then incubated for 24 h with shaking at 28 °C. An aliquot of 1 mL was taken from each sample and centrifuged at 10,000× *g* for 10 min. The pellet was collected and dissolved in 1 mL of DMSO. Then, the mixture was centrifuged at 10,000× *g* for 10 min after sufficient mixing. The absorbance of the supernatant containing violacein pigment was measured at 575 nm by the MK3 microcoder (Thermo Scientific, Waltham, MA, USA) [27]. To verify that carvacrol cannot degrade C4-HSL, violacein production was determined by adding *C. violaceum* CV026 in the presence of carvacrol and C4-HSL. All assays were performed in triplicate.

### 2.4. Determination AHL in A. hydrophila NJ-35

The synthesis of AHL in *A. hydrophila* NJ-35 was verified using an agar plate well diffusion assay [28]. Briefly, LB agar was poured into a sterilized culture dish following the overnight incubation of *C. violaceum* CV026. Next, 200 μL of cultivated bacteria was spread over the solidified LB agar plate followed by the punching of various holes using a cork borer of 10 mm diameter. The supernatant of *A. hydrophila* NJ-35 cultured to the late logarithmic growth stage was centrifuged and added into the experiment group holes (200 μL). The plates were then incubated at 28 °C for 24 h. The supernatant of *C. violaceum* CV026 served as the negative control, while the 40 μg/mL C4-HSL (solvent: sterile water) served as the positive control. The presence of AHL in the target strain was confirmed by the development of purple pigmentation due to violacein production by *C. violaceum* CV026.

### 2.5. Qualitative Effect of Carvacrol Concentration on AHLs Production of A. hydrophila NJ-35

The effect of the carvacrol concentration on AHLs production was observed. Briefly, *A. hydrophila* NJ-35 was inoculated with 1 × 10^8^ CFU/mL and diluted into LB (1:100 *v*/*v*) broth containing different concentrations of carvacrol (1/2 MIC = 62.5 μg/mL, 1/4 MIC = 31.25 μg/mL, 1/8 MIC = 15.625 μg/mL, and 1/16 MIC = 7.8125 μg/mL), while a negative control was treated with 1% DMSO. The preparation of an agar plate was similar to that described in Section 2.4. After incubation overnight on a shaker at 28 °C, 200 μL culture was spread in the agar plate. The plates were then incubated at 28 °C for 24 h. The effect of the carvacrol concentration on AHLs production was determined by the diameter of the violacein zones. The experiment was performed in triplicate, with three parallels in each group.

In order to further determine the content of AHL, a liquid–liquid extraction method was used to extract the cultures mentioned above [29]. After incubation overnight on a shaker at 28 °C, the cultures were grown to an OD_600_ = 1.0 and then subjected to centrifugation (4 °C, 10,000 r/min, 5 min); the supernatant was then mixed with an equal volume of ethyl acetate. After centrifugation at 10,000 r/min for 10 min, the upper organic phase was obtained and evaporated to 1 mL using a water bath (37 °C). The crude AHLs extract was stored at −80 °C. The crude AHL extract (40 μL) was added to the punched holes and incubated at 28 °C for 24 h. The diameters of the AHL-induced zone around the wells were measured using vernier calipers. The experiment was performed in triplicate, with three parallels in each group.

### 2.6. RT-qPCR Analysis

The effect of carvacrol on QS gene expression was assessed using a CFX96 RT-qPCR (Bio-Rad, Hercules, CA, USA). Briefly, *A. hydrophila* was inoculated into fresh LB broth with varying concentrations of carvacrol (1/16 MIC, 1/4 MIC) at 28 °C for 20 h, and 1% DMSO was used as a negative control. Total RNA was extracted following the instructions and guidelines of the RNAiso Plus kit (Takara, Daling, China). RNA quantities and concentrations were determined using a Nanodrop 2000 Spectrophotometer (Thermo Scientific, Waltham, MA, USA). Next, the cDNA was synthesized by reverse transcription using Hiscript RT supermix for qPCR with a gDNA wiper (Vazyme, Nanjing, China). Real-time PCR was performed using SYBR green real-time PCR mix (Bio-Rad) on a CFX real-time PCR detection system (Bio-Rad, Hercules, CA, USA) to detect the transcription level of the genes. Target gene expressions were calculated via the Pfaffl’s mathematical model [30], and *rpoB* was used as the reference housekeeping gene. All assays were carried out in triplicate. The gene-specific primers used in this study are listed in Table 1.

### 2.7. Molecular Docking Analysis

AhyI (GenBank: ABD59318.1) and AhyR (GenBank: ABD59317.1) of *A. hydrophila* were retrieved from the National Centre for Biotechnology Information (NCBI) protein database for further computer-simulated analysis. The three-dimensional (3D) structures of the homologous protein were predicted and assessed by SWISS-MODEL (https://swissmodel.expasy.org, accessed on 11 April 2023). The whole AhyI/R molecule associated with QS was identified as the binding site residue. Subsequently, the molecular docking simulation of carvacrol to AhyR or AhyI was performed with in silico analysis conducted by Autodock 4.2.3. The Lamarckian genetic algorithm implemented in Autodock was utilized to predict the possible conformations of the ligand binding to the AhyI/R proteins. Optimal docking was selected based on the calculated binding free energy.

### 2.8. Statistical Analysis

The data were expressed as the mean ± standard error of three independent experiments. A one-way ANOVA with Tukey’s multiple comparison post-test was conducted using SPSS 20.0 software (IBM Corp., New York, NY, USA) to analyze statistical differences among the groups.

## 3. Results

### 3.1. Antibacterial Activity of Carvacrol

To understand the sensitivity of *A. hydrophila* and *C. violaceum* CV026 to carvacrol and determine the concentration for carvacrol treatment, we measured the MIC of carvacrol against both bacterial strains. As shown in Figure 1, the MICs of carvacrol against *A. hydrophila* NJ-35 and *C. violaceum* CV026 determined in this study were 125 μg/mL and 62.5 μg/mL, respectively. The following experiments were conducted using carvacrol at concentrations below the MIC.

### 3.2. Effect of Carvacrol on the Production of Violacein in C. violaceum CV026

The effect of carvacrol concentration on the production of QS-mediated violacein by an indicator strain (*C. violaceum* CV026) was measured. Obviously, compared with the concentration of carvacrol required to inhibit the growth of bacteria, the production of violacein decreased significantly when the carvacrol dose increased (*p* < 0.05), and for the growth of bacteria there was no obvious difference until the dose of carvacrol increased to 1/4 MIC (Figure 2A). Moreover, the relationship of carvacrol and C4-HSL was investigated, as shown in Figure 2B. There was no obvious difference in the biological activity after the addition of exogenous C4-HSL, indicating that carvacrol could not degrade C4-HSL. Therefore, at 1/4 MIC concentration, carvacrol can effectively inhibit the production of violacein without affecting the biological activity of *C. violaceum* CV026, indicating that carvacrol can inhibit the QS system of *C. violaceum* CV026.

### 3.3. The Presence of AHLs in A. hydrophila NJ-35

As shown in Figure 3, the purple color could be observed in the positive control and experiment group, but no purple color was observed with the negative control. Compared with the negative control, the presence of C4-HSL in the positive control group could induce *C. violaceum* CV026 to produce violacein. In addition, the observation of purple in the experiment group demonstrated that the supernatant of *A. hydrophila* NJ-35 existed violacein, which indicated that *A. hydrophila* NJ-35 can synthesize and secrete the signal molecule AHLs.

### 3.4. Effect of Carvacrol on AHLs Production in A. hydrophila NJ-35

A qualitative analysis was conducted to investigate the impact of carvacrol on the AHLs production of *A. hydrophila* NJ-35. The results showed anti-QS properties of carvacrol, as evidenced by a reduction in the diameter of the violacein zones (Figure 4). No significant difference in violacein zones was observed between 1/8 MIC and 1/16 MIC when compared to the negative control (DMSO) (*p* > 0.05). However, a significant reduction in the average diameter of the violacein zones was observed at 1/2 MIC and 1/4 MIC carvacrol (*p* < 0.05). Therefore, when the concentration of carvacrol is low, it does not affect the production of violacein. Only when the concentration of carvacrol reaches a level (1/4 MIC) will it inhibit the production of violacein.

Moreover, the inhibitory effects of carvacrol on AHLs were validated and quantified through the spectrophotometric measurement of violacein concentrations. As depicted in Figure 5, no significant differences were observed between the 1/8 MIC and 1/16 MIC groups and the control group with respect to AHLs production (*p* > 0.05). However, the production of violacein was noticeably reduced in the 1/4 MIC group treated with carvacrol compared to the control group (DMSO) (*p* < 0.05), demonstrating the ability of carvacrol to impede the production of AHLs signaling molecules in *A. hydrophila*. In addition, when the concentration of carvacrol reaches 1/2 MIC, it not only enhances its inhibitory effect on the production of violacein but also affects the growth of *A. hydrophila* due to its toxicity. Therefore, the optimal concentration of carvacrol is chosen to be 1/4 MIC.

### 3.5. Effect of Carvacrol on QS-Associated Genes of A. hydrophila

The impact of carvacrol on the expression of the AHL synthase genes (*ahyI/R*) was assessed using quantitative PCR. As shown in Figure 6, the group treated with 1/16 MIC of carvacrol showed a 0.90-fold change in the expression level of *ahyI* gene and a 0.95-fold change in the expression level of *ahyR* gene. However, the difference between the 1/16 MIC group and control group (DMSO) was not obvious (*p* > 0.05). Conversely, the group treated with carvacrol at 1/4 MIC exhibited a 0.41-fold change in the expression level of *ahyI* gene and a 0.85-fold change in the expression level of *ahyR* gene, and the difference was significant compared to the 1/16 MIC group.

### 3.6. Molecular Docking

The interaction of carvacrol and C4-HSL into the AhyI/R receptor active site was investigated by molecular docking analysis. As shown in Figure 7A,B, carvacrol interacted with the Ser145 residue of AhyI protein and His126 residue of AhyR protein via hydrogen bonds (yellow dashed lines). The binding free energy of AhyI/R to carvacrol was −5.51 kcal/mol and −4.94 Kcal/mol, respectively (Table 2). In addition, C4-HSL interacted with the His126, Pro124, and Asn161 residues of AhyR protein via the hydrogen bonds, as illustrated in Figure 7C, with a binding free energy of −5.58 Kcal/mol (Table 2). Although receptor protein AhyR is more likely to bind with C4-HSL due to their lower binding free energy, the binding free energy between carvacrol and receptor protein AhyR is close to that between AhyR and C4-HSL. Therefore, carvacrol can competitively bind with the receptor protein AhyR, reducing the formation of AhyR-AHL complexes and inhibiting the QS system of *A. hydrophila* [33].

## 4. Discussion

The pathogenicity of *A. hydrophila* is mainly due to virulence traits, such as extracellular protease, adhesion, and exotoxin mediated by AHL-mediated QS [34]. Therefore, developing a natural QS inhibitor to interfere with AHLs production in *A. hydrophila* is a promising bacteriostatic strategy. Compared with the traditional germicidal and bacteriostatic strategies, the QS inhibitor cannot only eliminate bacterial pathogenicity or improve their sensitivity to antibiotics without killing microorganisms, but also prevent the formation of antibiotic-resistance bacteria [35]. The bactericidal activity of carvacrol depends on its concentration and exposure time. Certain concentrations of carvacrol corresponding to MIC provoke the death of different bacterial cells. Carvacrol exerts its bactericidal ability by altering the permeability of the cell membrane to cations such as H^+^ and K^+^. The disruption of ion gradients leads to damage in crucial cellular processes, ultimately resulting in cell death [36]. Carvacrol concentrations lower than MIC cannot result in the death of cells but can reduce their pathogenicity by inhibiting their QS. That was shown for *P. aeruginosa*, *Salmonella* enterica subsp, Typhimurium DT104, and *S. aureus* [22,37,38]. Our research aimed to used carvacrol as a natural QS inhibitor which can be applied to attenuate the pathogenicity of *A. hydrophila* NJ-35. Definite concentrations resulting in such an effect were revealed.

In order to confirm that carvacrol has the function of QS inhibition, we used the QS model strain *C. violaceum* CV026, which can investigate the QS system by monitoring the production of violacein. The QS of *C. violaceum* CV026 depends on the presence of exogenous AHLs due to the lack of the LuxI-type synthase in *C. violaceum* CV026 [39]. Hence, the production of violacein could quantify the presence of exogenous AHLs in the environment and evaluate the anti-QS activity of phytochemicals. Our results showed that the production of violacein secreted by *C. violaceum* CV026 decreased by 62.46% after 1/4 MIC carvacrol treatment. In addition, 1/4 MIC carvacrol could not interfere with bacterial growth (Figure 2A), which was consistent with the result reported previously [40]. Moreover, the presence of carvacrol could not degrade the signal molecular (Figure 2B). Therefore, carvacrol has the potential to inhibit the synthesis of the signal molecules of AHL in the QS system of *A. hydrophila* NJ-35.

The content of the signal molecules of AHL was detected to determine the effect of carvacrol on the pathogenicity of *A. hydrophila*, which was regulated by the QS system because *A. hydrophila* can produce signal molecules of AHL by *ahyI* and *ahyR* genes [41,42,43]. Therefore, through an agar plate well diffusion assay, we found that the hole with *A. hydrophila* NJ-35 was purple, indicating that *A. hydrophila* NJ-35 could produce signal molecules of AHL (Figure 3). The average diameter of the violacein zones with 1/2 MIC and 1/4 MIC were less than that 1/8 MIC and 1/16 MIC obviously. In addition, the production of violacein decreased by 36.01% after 1/4 MIC carvacrol treatment, and the growth of *A. hydrophila* NJ-35 was not affected (Figure 4 and Figure 5). Therefore, 1/4 MIC carvacrol was chosen for further research.

Moreover, AhyR/I is responsible for regulating the AI-1 QS system of *A. hydrophila*, in which AhyI is responsible for the synthesis of signal molecules AHLs, and AhyR recognition signal molecules and feedback regulation activates downstream virulence factors [44]. Therefore, the inhibition of carvacrol on the QS of *A. hydrophila* NJ-35 was further characterized by detecting the expression of *ahyI* (AHL synthase) and *ahyR* (receptor protein) genes [45]. The results demonstrated that the expression level of *ahyI* and *ahyR* was down-regulated obviously after being treated by carvacrol (Figure 6), indicating that carvacrol could interfere with the expression of these genes. The down-regulation of these genes expression levels directly affects the synthesis of the signal molecule AHL and the binding of the signal molecule to the receptor protein.

To reveal the mechanism of how carvacrol mediated the inhibition of the QS system of *A. hydrophila* NJ-35, molecular docking was analyzed. We found that carvacrol can be effectively combined with Ser145 of AhyI and His126 of AhyR via hydrogen bond, and the binding free energy is −5.51 kcal/mol and −4.94 kcal/mol, respectively. In addition, the signal molecules of AHL (C4-HSL) can be combined with His126, Pro124, and Asn161 of AhyR via hydrogen bonding, and the binding free energy is −5.58 kcal/mol. Obviously, carvacrol and AHL can form hydrogen bonds at the same site, His126, which allows carvacrol to competitively bind to the AhyR and decreases the formation of AhyR-AHL composite, thus alleviating the pathogenicity of *A. hydrophila* NJ-35.

Based on the results detected in this study, the possible action of carvacrol reducing the pathogenicity of *A. hydrophila* NJ-35 was proposed (Figure 8). By exploring the effect of carvacrol on the production of violacein secreted by *C. violaceum* CV026, we found that carvacrol used QS inhibition, decreasing the synthesis of QS signal molecules of AHL in *A. hydrophila* NJ-35. On the one hand, carvacrol could decrease the production of the QS signal molecules of AHL by reducing the expression of *ahyI* gene. On the other hand, carvacrol could down-regulate the expression level of *ahyR*, thus decreasing the synthesis of AhyR receptor. Finally, the reduction in the signal molecules of AHL and receptor AhyR could affect the formation of AhyR-AHL complexes, resulting in the decrease in virulence factors, and thus mitigate the pathogenicity of *A. hydrophila* NJ-35.

## 5. Conclusions

In conclusion, we have confirmed that carvacrol can be used as a natural QS inhibitor, and we applied it to inhibit the QS system of *A. hydrophila* in aquaculture. Furthermore, we have elucidated the interaction between carvacrol and QS, and proposed a QS inhibition for decreasing the pathogenicity of *A. hydrophila*. These findings demonstrate that the natural QS inhibitor carvacrol holds great potential for replacing antibiotics due to its effective QS inhibitory abilities. This study has laid the theoretical foundation and basis for the development of drugs using QS inhibition to treat fish diseases.

## Figures and Tables

**Figure 1 microorganisms-11-02027-f001:**
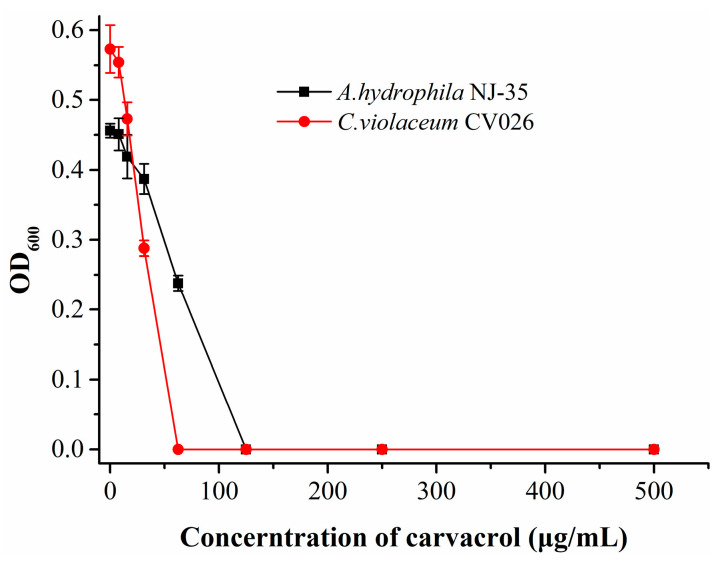
MIC of carvacrol against *A. hydrophila* NJ-35 and *C. violaceum* CV026.

**Figure 2 microorganisms-11-02027-f002:**
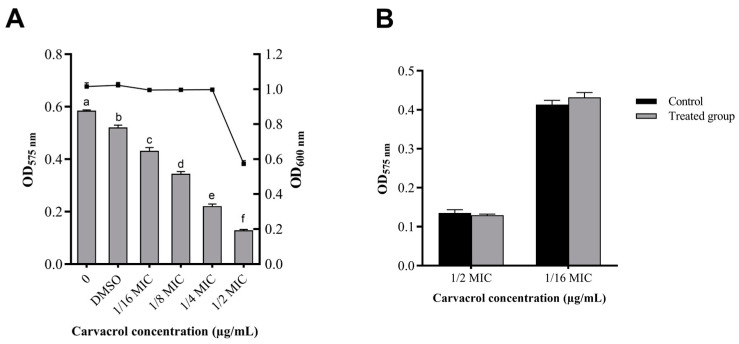
Violacein production in *C. violaceum* CV026 after carvacrol treatment. (**A**) The effect of carvacrol on the production of violacein: the column shows the production of violacein and the dotted line shows the biomass of *C. violaceum* CV026. (**B**) The relationship of carvacrol and C4-HSL. a–f: Values with different letters are significantly different (*p* < 0.05), while those with similar letters are not.

**Figure 3 microorganisms-11-02027-f003:**
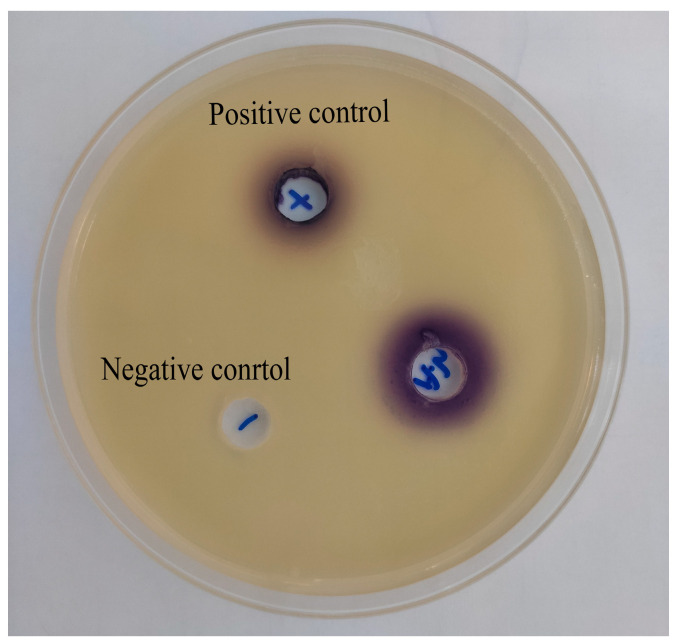
Detection of AHLs in *A. hydrophila* NJ-35.

**Figure 4 microorganisms-11-02027-f004:**
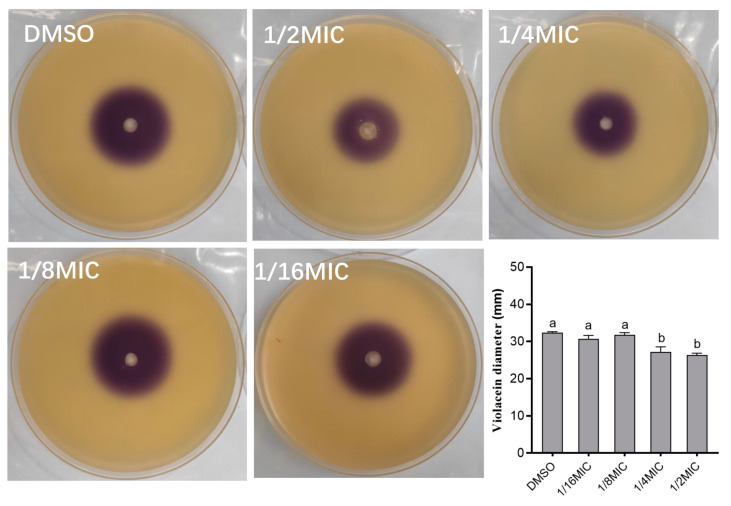
Qualitative effect of carvacrol on AHLs production of *A. hydrophila* NJ-35. a,b: Values with different letters are significantly different (*p* < 0.05), while those with similar letters are not.

**Figure 5 microorganisms-11-02027-f005:**
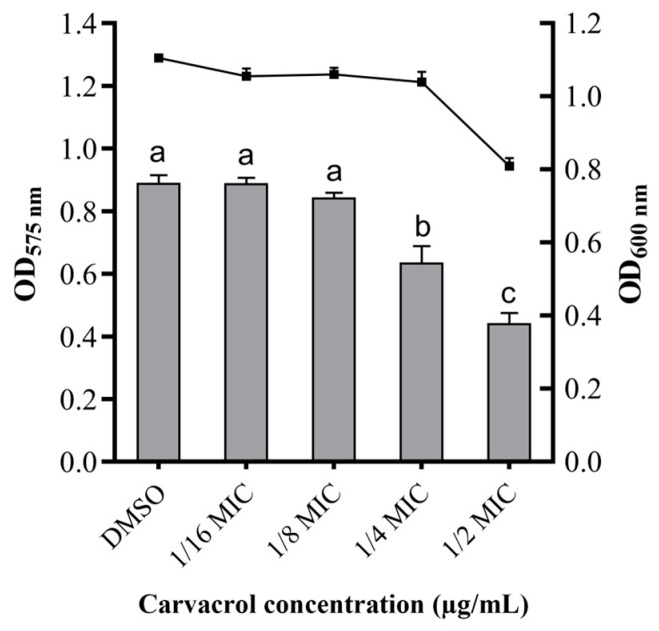
Quantitative effect of carvacrol on AHLs production of *A. hydrophila*. a–c: Values with different letters are significantly different (*p* < 0.05), while those with similar letters are not. The column shows the production of AHLs, and the dotted line shows the biomass of *A. hydrophila*.

**Figure 6 microorganisms-11-02027-f006:**
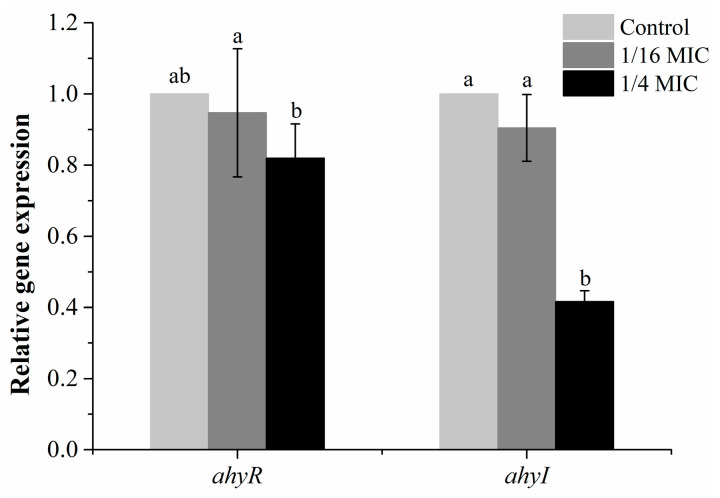
Effect of carvacrol on QS-related gene expression in *A. hydrophila.* a,b: Values with different letters are significantly different (*p* < 0.05), while those with similar letters are not. Data are shown as the value of the carvacrol-treated (1/4 MIC, 1/16 MIC) group normalized to the DMSO group.

**Figure 7 microorganisms-11-02027-f007:**
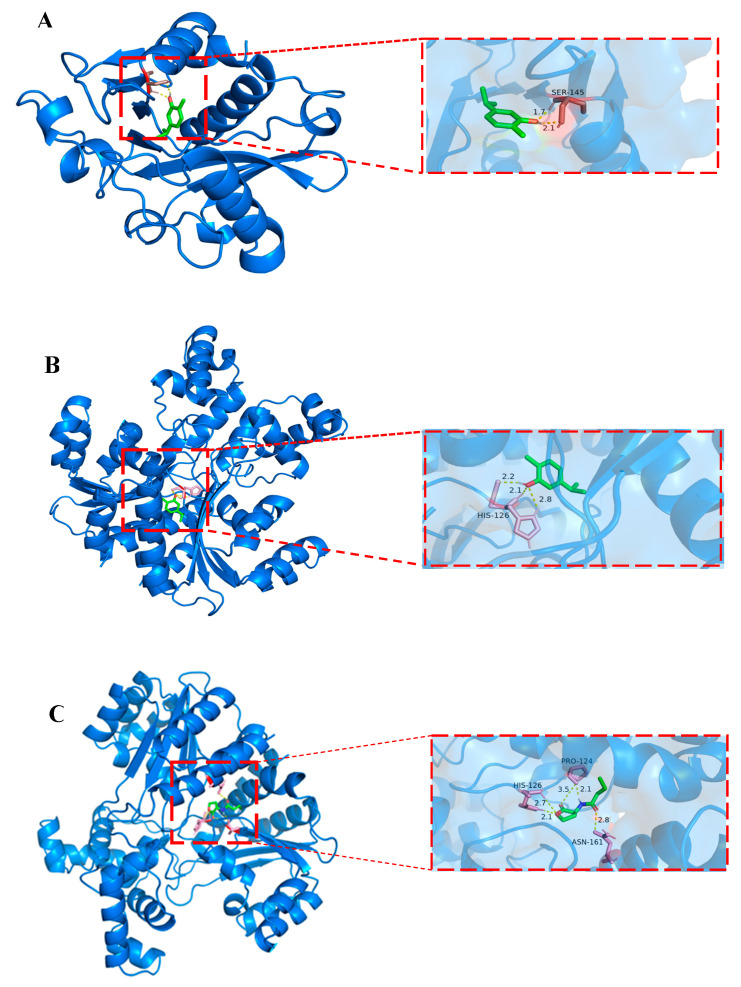
Molecular docking analysis. (**A**) Carvacrol combined with AhyI protein; (**B**) carvacrol combined with AhyR protein; (**C**) C4-HSL combined with AhyR protein.

**Figure 8 microorganisms-11-02027-f008:**
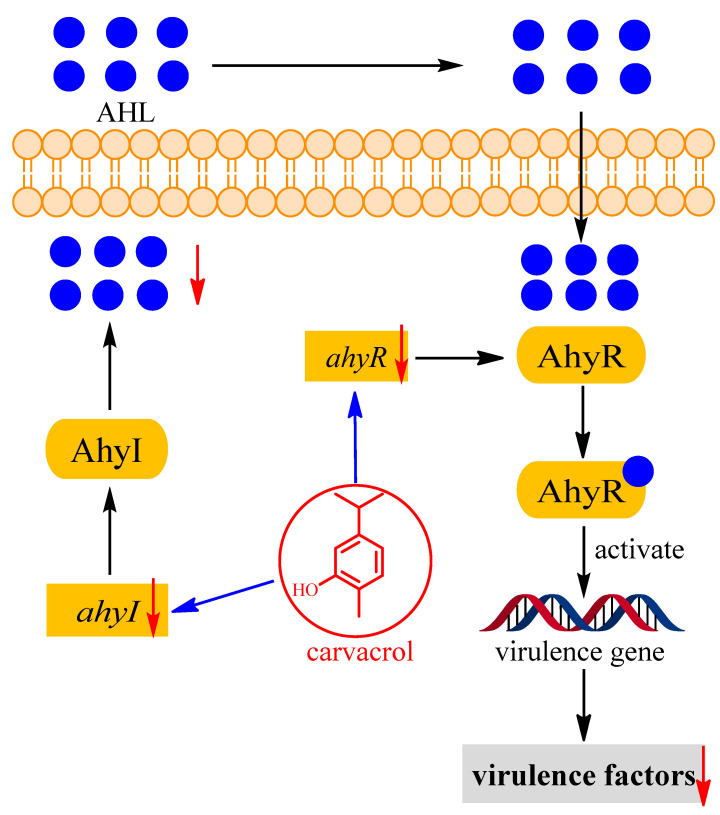
The possible action of carvacrol alleviates the pathogenicity of *A. hydrophila* NJ-35.

**Table 1 microorganisms-11-02027-t001:** Primers used for quantitative PCR.

Primer	Sequence (5′→3′)	Reference
*ahyR*-F	TCTTGACGTGATGGGGTTGG	[31]
*ahyR*-R	GGCGGTGATGAACGACAGTA	
*ahyI*-F	CAGATGGGAGGTAGAAAACGAG	[31]
*ahyI*-R	TGGGTATCAGGGGTATCGAAA	
*ropB*-F	ACCGACGAAGTGGACTATCT	[32]
*ropB*-R	CGGCGTTCATAAAGGTGGAT	

**Table 2 microorganisms-11-02027-t002:** Molecular docking results of the interaction of carvacrol and AhyI/R.

Compound	Protein	Binding Free Energy (Kcal/mol)	Hydrogen Bond Interacting Residues
Carvacrol	AhyI	−5.51	Ser145
	AhyR	−4.94	His126
C4-HSL	AhyR	−5.58	His126 Pro124 Asn161

## Data Availability

The raw data supporting the conclusions of this article will be made available by the authors, without undue reservation.

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
