# Peer review of "Carvacrol Inhibits Quorum Sensing in Opportunistic Bacterium Aeromonas hydrophila"

_microorganisms, 2023, doi:10.3390/microorganisms11082027_

Round 1
Reviewer 1 Report
I have reviewed the manuscript, and here are the comments:-
English needs to improve; there are some sentences and parts of the manuscript that are difficult to understand
Title: bacterium A. hydrophila: It is known to be bacteria. Please change the address to Carvacrol inhibits quorum sensing in opportunistic Aeromonas hydrophila
Many abbreviations in the manuscript need to be cleared, and add more results in the abstract.
Clear the aim of the study.
Please arrange all keywords in alphabetical order
Provide the origin and model of all devices.
Many methods need references.
The materials and methods are difficult to follow in many cases. Extensive revisions are required. The authors should read the manuscript carefully and provide more details in any analysis. In addition, provide the proper reference.
The discussion is old and needs to be updated, and the references don’t follow the journal instruction.
To improve the quality of the paper, update the reference list by adding 2022 and 2023 references.
The conclusion shouldn’t contain results, check and rewrite this part.
Please follow the authors' instructions on how they write the reference in the list. Why are you capitalizing the first letter of every word? Please see the journal style. For references about textbooks, please add the page numbers of the textbook. Also, please add the city of the publisher.
Enhance the resolution of Figures 1,4 and 5.
English needs to improve; there are some sentences and parts of the manuscript that are difficult to understand
Author Response
Dear editor,
Thank you for considering our manuscript entitled “Carvacrol inhibits quorum sensing in opportunistic bacterium A. hydrophila” for the publication in Microorganisms. We greatly appreciate the comments and suggestions from the referees and the editors, which inspired us to further explore our investigation. As to some questions raised by the reviewers, we have the following replies and the replies were highlighted in blue. Notably, the title has been changed to “Carvacrol inhibits quorum sensing in opportunistic bacterium Aeromonas hydrophila” according to the suggestions of reviewer 1 and reviewer 2. We also checked the grammar and typos carefully and revised our manuscript according to the reviewers’ opinions. The revised manuscript with changes using yellow highlights for easy identification was also prepared.
To Reviewer 1:
Q1: English needs to improve; there are some sentences and parts of the manuscript that are difficult to understand
A1: Thanks for your advice. We have improved the language of the manuscript carefully to avoid grammatical, and bibliographic errors.
Q2: Title: bacterium A. hydrophila: It is known to be bacteria. Please change the address to Carvacrol inhibits quorum sensing in opportunistic Aeromonas hydrophila
A2: Thanks for the reviewer’s suggestion on the title. We have changed the title as follows: Carvacrol inhibits quorum sensing in opportunistic bacterium Aeromonas hydrophila
Q3: Many abbreviations in the manuscript need to be cleared, and add more results in the abstract.
A3: Thanks for your suggestion to strengthen the importance of our work. We have cleared the abbreviations that appeared only once. In addition, we have added the detailed results data of the carvacrol concerntration and efficiency in the abstract.
Q4: Clear the aim of the study.
A4: Thanks for your advice. We have cleared the aim of the study.
Q5: Please arrange all keywords in alphabetical order
A4: Thanks for your advice. We have arrange all keywords in alphabetical order.
Q6: Provide the origin and model of all devices.
A6: Thanks for your advice. We have added the origin and model of all devices.
Q7: Many methods need references.
A7: Thanks for your advice. We have added the references in the mehods
Q8: The materials and methods are difficult to follow in many cases. Extensive revisions are required. The authors should read the manuscript carefully and provide more details in any analysis. In addition, provide the proper reference.
A8: Thanks for your advice. According to the reviewer’s suggestion, we have added more references in the section of materials and mehods. In addition, we have revised some details of the method to improve the readability of the article.
Q9: The discussion is old and needs to be updated, and the references don’t follow the journal instruction.
A9: Thanks for the reviewer’s suggestion on the improvement of the article. We have revised the discussion in the text.
Q10: To improve the quality of the paper, update the reference list by adding 2022 and 2023 references.
A10: Thanks for your advice. We have replaced the early published references with the latest published references.
Q11: The conclusion shouldn’t contain results, check and rewrite this part.
A11: Thanks for the reviewer’s suggestion on the improvement of the article. We have rewrite the conclusion as follows.
In conclusion, we have confirmed that carvacrol can be used as natural QS inhibitor and applied it to inhibit the QS system of A. hydrophila in aquaculture. Furthermore, we have elucidated the interaction between carvacrol and QS, and proposed a QS inhibition mechanism for decreasing the pathogenicity of A. hydrophila. These findings demonstrate that natural QS inhibitors carvacrol hold great potential for replacing antibiotics due to their effective QS inhibitory abilities. This study has laid the theoretical foundation and basis for the development of drugs using QS inhibition mechanism to treat fish diseases.
Q12: Please follow the authors' instructions on how they write the reference in the list. Why are you capitalizing the first letter of every word? Please see the journal style. For references about textbooks, please add the page numbers of the textbook. Also, please add the city of the publisher.
A12: Thanks for your advice. We have carefully checked the format of references and revised them.
Q13: Enhance the resolution of Figures 1,4 and 5.
A13: Thanks for your advice. We have enhanced the resolution of Figuers 1, 4 and 5.
Q14: English needs to improve; there are some sentences and parts of the manuscript that are difficult to understand.
A14: Thanks for your advice. We have improved the language of the manuscript carefully to avoid grammatical, and bibliographic errors.

Reviewer 2 Report
Article title and Abstract
It is necessary to give the full names of microorganisms both in the title of the article and in the Abstract in the places where they are remembered for the first time in the text. So it is necessary to replace A. hydrophila with Aeromonas hydrophila and C. violaceum with Chromobacterium violaceum.
Introduction
Lines 54-56: Generally speaking, each type of bacteria can only recognize the AHL molecule that they produce, hence the AHL system exhibits species-specificity [13].
Lines 67-69: There is following text: “In addition, our previous research has demonstrated that carvacrol, the principal component of oregano essential oil extracted from the plant Origanum vulgare, can effectively inhibit the growth and virulence of A. hydrophila at sub-inhibitory concentrations [20, 21].” According to the text, in references 20 and 21 should present the previous results of the authors of the same article ("our previous research"), but in article 20 they are not the authors. In this regard, the phrase in the text should be corrected.
All the names of the authors of the articles cited in the text (for example, Cosa et al (line 65), Tapia-Rodriguez et al (line 71), Wang et al (line 146), etc.) should be removed from the text by rearranging the phrases.
Text on lines 75-84 ("In this study, we demonstrated that carvacrol acted as a natural QS inhibitor using Chromobacterium violaceum CV026, which was a biosensor bacterium strain and produced violacein in the presence of an exogenously supplied short chain AHL, and further extended its application to reduce the toxicity of A. hydrophila. It can not only reduce the concentration of signal molecule AHL in QS system of A. hydrophila, but also down-regulated the expression of key genes ahyR/I, thus inhibiting QS system and reducing the pathogenic ability of A. hydrophila. Moreover, carvacrol can be effectively combined with Ser145 of AhyI and His126 of AhyR via hydrogen bond. Finally, the mechanism of carvacrol reducing pathogenicity of A. hydrophila was proposed. This work provided the theoretical foundation for novel antibacterial therapy in aquaculture.") should be modified and presented not as the results already obtained, but as the purpose of the study.
Results
The part 3.1 is too short and should be combined with part 3.2, after that the title of part 3.2 should be modified.
Lines 186-187: There I following text: “The MIC of carvacrol against A. hydrophila NJ-35 and C. violaceum CV026 determined in this study were 125 μg/mL and 62.5 μg/mL, respectively.” There are no experimental data confirming these results in this article. For A. hydrophila NJ-35, the value of MIC was determined by the authors and presented in another work published by them (Wang, J.; Qin, T.; Chen, K.; Pan, L.; Xie, J.; Xi, B. Antimicrobial and Antivirulence Activities of Carvacrol against Pathogenic Aeromonas hydrophila. Microorganisms 2022, 10, 2170. https://doi.org/10.3390/microorganisms10112170). There is no link to this article at this point in the text. It needs to be added. There are no data on the definition of MIC of carvacrol against C. violaceum CV026 in the article, and they must be presented in this article.
Figure 1 and Figure 4: The graph shows columns and a line with dots. It is not clear what applies to what.
Figure 2: a “positive control" is presented, where an additional AHL molecule is added (in addition to the AHL-synthesizing bacteria A.hydrophila themselves), which in theory should lead to even greater synthesis of violacein, but according to the figure it does not work that way. Or if there is only an AHL molecule in the “positive control”, without A.hydrophila bacteria, then this should be more clearly spelled out in the methods so that there is no confusion.
Figure 5: According to the data presented, 1/16 of the MIC concentration leads to an increase in the expression of ahyI/R genes. How is that, why? An explanation of the results is needed.
Figure 6: It would be necessary to add an explanation of how carvacrol interacts with AhyI/R proteins, in particular, with the AGL-synthesizing protein AhyI.
Figure 7: "the mechanism by which pathogenicity is inhibited" sounds too loud. There is no mechanism, except only the intended path.
General recommendation to the article:
- I propose in the figures to represent carvacrol concentrations along the abscissa axis not in the form of parts from MIC, but in the form of real concentrations that the authors know. In this case, the data of this article will be easier to compare with the data of other researchers, in particular, who established already in 2014 that the natural antimicrobial carvacrol inhibits quorum sensing in Chromobacterium vilaceum and reduces bacterial biofilm formation at sub-lethal concentrations (Burt SA, Ojo-Fakunle VT, Woertman J, Veldhuizen EJ. The natural antimicrobial carvacrol inhibits quorum sensing in Chromobacterium violaceum and reduces bacterial biofilm formation at sub-lethal concentrations. PLoS One. 2014 9(4):e93414. doi: 10.1371/journal.pone.0093414.)
- I suggest that the authors discuss in the article the previously known mechanism of action of carvacrol on bacterial cells (Carvacrol interacts with the membranes of cells by changing its permeability for cations like H(+) and K(+). The dissipation of ion gradients leads to impairment of essential processes in the cell and finally to cell death. [Ultee A, Kets EP, Smid EJ. Mechanisms of action of carvacrol on the food-borne pathogen Bacillus cereus. Appl Environ Microbiol. 1999 Oct;65(10):4606-10. doi: 10.1128/AEM.65.10.4606-4610.1999.]) This mechanism is more suitable even for gram-negative cells, with which the authors work, than with gram-positive cells, on which this effect of carvacrol on cell membranes was established). The authors do not mention it in any way in their article, offering their own, fundamentally different way of exposure to carvacrol. This is not good, because this is the novelty of the article, and it can be increased with this discussion. Now the novelty is not clear, so it will be useful to emphasize this at the end of the article, taking into account the references (including previous paper of same authors) I mentioned in the review.
Author Response
Dear editor,
Thank you for considering our manuscript entitled “Carvacrol inhibits quorum sensing in opportunistic bacterium A. hydrophila” for the publication in Microorganisms. We greatly appreciate the comments and suggestions from the referees and the editors, which inspired us to further explore our investigation. As to some questions raised by the reviewers, we have the following replies and the replies were highlighted in blue. Notably, the title has been changed to “Carvacrol inhibits quorum sensing in opportunistic bacterium Aeromonas hydrophila” according to the suggestions of reviewer 1 and reviewer 2. We also checked the grammar and typos carefully and revised our manuscript according to the reviewers’ opinions. The revised manuscript with changes using yellow highlights for easy identification was also prepared.
To Reviewer 2:
Q1: It is necessary to give the full names of microorganisms both in the title of the article and in the Abstract in the places where they are remembered for the first time in the text. So it is necessary to replace A. hydrophila with Aeromonas hydrophila and C. Violaceum with Chromobacterium violaceum.
A1: Thanks for your advice. We have replaced A. hydrophila with Aeromonas hydrophila and C. Violaceum with Chromobacterium violaceum when A. hydrophila and C. Violaceum first appear in the text.
Q2: Lines 54-56: Generally speaking, each type of bacteria can only recognize the AHL molecule that they produce, hence the AHL system exhibits species-specificity [13].
A2: Thanks for your advice. According to the previous setence “ the AHL molecules produced by each type of bacteria are different, mainly due to variations in the length of the fatty acid chain”, we changed it as “Therefore, each type of bacteria can only recognize the AHL molecule that they produce, hence the AHL system exhibits species-specificity”.
Q3: Lines 67-69: There is following text: “In addition, our previous research has demonstrated that carvacrol, the principal component of oregano essential oil extracted from the plant Origanum vulgare, can effectively inhibit the growth and virulence of A. hydrophila at sub-inhibitory concentrations [20, 21].” According to the text, in references 20 and 21 should present the previous results of the authors of the same article ("our previous research"), but in article 20 they are not the authors. In this regard, the phrase in the text should be corrected.
A3: Thanks for your advice. We have revised the sentence in the text.
Q4: All the names of the authors of the articles cited in the text (for example, Cosa et al (line 65), Tapia-Rodriguez et al (line 71), Wang et al (line 146), etc.) should be removed from the text by rearranging the phrases.
A4: Thanks for your advice. We have removed all the names of the authors of the articles cited in the text.
Q5: Text on lines 75-84 ("In this study, we demonstrated that carvacrol acted as a natural QS inhibitor using Chromobacterium violaceum CV026, which was a biosensor bacterium strain and produced violacein in the presence of an exogenously supplied short chain AHL, and further extended its application to reduce the toxicity of A. hydrophila. It can not only reduce the concentration of signal molecule AHL in QS system of A. hydrophila, but also down-regulated the expression of key genes ahyR/I, thus inhibiting QS system and reducing the pathogenic ability of A. hydrophila. Moreover, carvacrol can be effectively combined with Ser145 of AhyI and His126 of AhyR via hydrogen bond. Finally, the mechanism of carvacrol reducing pathogenicity of A. hydrophila was proposed. This work provided the theoretical foundation for novel antibacterial therapy in aquaculture.") should be modified and presented not as the results already obtained, but as the purpose of the study.
A5: Thanks for your advice. According to the reviewer’s suggestion, the revised sentences have been attached as follows.
In this study, we demonstrated that carvacrol acted as a natural QS inhibitor using Chromobacterium violaceum CV026, which was a biosensor bacterium strain and produced violacein in the presence of an exogenously supplied short chain AHL, and further extended its application to reduce the toxicity of A. hydrophila. Moreover, to investigate the mechanism of carvacrol reducing pathogenicity of A. hydrophila by carvacrol we studied the concentration changes of AHL signal molecules in A. hydrophila QS system, as well as the expression of related genes. Additionally, we performed a simulation analysis to examine the binding mode between carvacrol and the receptor protein, and calculated their binding free energy. This work provided the theoretical foundation for novel antibacterial therapy in aquaculture.
Q6: The part 3.1 is too short and should be combined with part 3.2, after that the title of part 3.2 should be modified.
A6: Thanks for your suggestions to improve the reliability and quality of the article. We have added some describtion in section 3.1 and the date of MIC of carvacrol against A. hydrophila NJ-35 and C. violaceum CV026.
Q7: Lines 186-187: There I following text: “The MIC of carvacrol against A. hydrophila NJ-35 and C. violaceum CV026 determined in this study were 125 μg/mL and 62.5 μg/mL, respectively.” There are no experimental data confirming these results in this article. For A. hydrophila NJ-35, the value of MIC was determined by the authors and presented in another work published by them (Wang, J.; Qin, T.; Chen, K.; Pan, L.; Xie, J.; Xi, B. Antimicrobial and Antivirulence Activities of Carvacrol against Pathogenic Aeromonas hydrophila. Microorganisms 2022, 10, 2170. https://doi.org/10.3390/microorganisms10112170). There is no link to this article at this point in the text. It needs to be added. There are no data on the definition of MIC of carvacrol against C. violaceum CV026 in the article, and they must be presented in this article.
A7: Thanks for your advice. We have added the date of MIC of carvacrol against A. hydrophila NJ-35 and C. violaceum CV026 as follows.
Fig. 1 MIC of carvacrol against A. hydrophila NJ-35 and C. violaceum CV026.
Q8: Figure 1 and Figure 4: The graph shows columns and a line with dots. It is not clear what applies to what.
A8: Thanks for your advice. We have explained their meaning in the text. The columns show the production of violacein and AHLs in Fig 1 and Fig 4, respectively. The line of dots show the biomass of C. violaceum CV026 and A.hydrophila in Fig 1 and Fig 4, respectively.
Q9: Figure 2: a “positive control" is presented, where an additional AHL molecule is added (in addition to the AHL-synthesizing bacteria A.hydrophila themselves), which in theory should lead to even greater synthesis of violacein, but according to the figure it does not work that way. Or if there is only an AHL molecule in the “positive control”, without A.hydrophila bacteria, then this should be more clearly spelled out in the methods so that there is no confusion.
A9: Thanks for the reviewer’s suggestion on the improvement of the article. In the positive group, there is only an AHL molecule in the “positive control”, without A.hydrophila bacteria. We have clearly spelled out in the methods.
Q10: Figure 5: According to the data presented, 1/16 of the MIC concentration leads to an increase in the expression of ahyI/R genes. How is that, why? An explanation of the results is needed.
A10: Thanks for your advice. Although the expression of ahyI/R genes were increased after 1/16 MIC treatment, there was no obvious significant difference compare to the control. However, we found it are unreasonable. Thus we carried out another group of experiments of the genes expression for 3 times. The revised Fig. 6 was attached as follows.
Figure 6. Effect of carvacrol on QS-related gene expression in A. hydrophila. Data are shown as the value of the carvacrol treated (1/4 MIC, 1/16 MIC) group normalized to the DMSO group.
Q11: Figure 6: It would be necessary to add an explanation of how carvacrol interacts with AhyI/R proteins, in particular, with the AGL-synthesizing protein AhyI.
A11: Thanks for the reviewer to raise an important point here to improve the importance of our manuscript. Here, carvacrol interacted with Ser145 residue of AhyI protein and His126 residue of AhyR protein via the hydrogen bonds (yellow dashed lines).
Q12: Figure 7: "the mechanism by which pathogenicity is inhibited" sounds too loud. There is no mechanism, except only the intended path.
A12: Thanks for your advice. The aim of our research is to reduce the pathogenicity of A.hydrophila by inhibiting its QS system via carvacrol. Therefore, we propose a QS inhibition mechanism to reduce the pathogenicity of A.hydrophila. On the one hand, carvacrol could decrease the production of QS signal molecular AHL by reducing the expression of ahyI gene. On the other hand, carvacrol could down-regulating the expression level of ahyR, thus decreasing the synthesis of AhyR receptor. Finally, the reduce of signal molecular AHL and receptor AhyR could affect the formation of AhyR-AHL complexes, resulting in the decrease of virulence factors, and thus mitigate the pathogenicity of A. hydrophila NJ-35.
Q13: I propose in the figures to represent carvacrol concentrations along the abscissa axis not in the form of parts from MIC, but in the form of real concentrations that the authors know. In this case, the data of this article will be easier to compare with the data of other researchers, in particular, who established already in 2014 that the natural antimicrobial carvacrol inhibits quorum sensing in Chromobacterium vilaceum and reduces bacterial biofilm formation at sub-lethal concentrations (Burt SA, Ojo-Fakunle VT, Woertman J, Veldhuizen EJ. The natural antimicrobial carvacrol inhibits quorum sensing in Chromobacterium violaceum and reduces bacterial biofilm formation at sub-lethal concentrations. PLoS One. 2014 9(4):e93414. doi: 10.1371/journal.pone.0093414.)
A13: Thanks for your advice. The MIC of carvacrol against A. hydrophila and C. violaceum is different. In order to clearly observe the inhibitory effects of carvacrol on the QS of these two different strains, we used the MIC instead of the actual concentrations. Additionally, we have added the real concentrations used in the section of materials and methods.
Q14: I suggest that the authors discuss in the article the previously known mechanism of action of carvacrol on bacterial cells (Carvacrol interacts with the membranes of cells by changing its permeability for cations like H(+) and K(+). The dissipation of ion gradients leads to impairment of essential processes in the cell and finally to cell death. [Ultee A, Kets EP, Smid EJ. Mechanisms of action of carvacrol on the food-borne pathogen Bacillus cereus. Appl Environ Microbiol. 1999 Oct;65(10):4606-10. doi: 10.1128/AEM.65.10.4606-4610.1999.]) This mechanism is more suitable even for gram-negative cells, with which the authors work, than with gram-positive cells, on which this effect of carvacrol on cell membranes was established). The authors do not mention it in any way in their article, offering their own, fundamentally different way of exposure to carvacrol. This is not good, because this is the novelty of the article, and it can be increased with this discussion. Now the novelty is not clear, so it will be useful to emphasize this at the end of the article, taking into account the references (including previous paper of same authors) I mentioned in the review.
A14: Thanks for your advice. We have added this description and cited this artical in the section of discussion as follows.
As a bactericidal compound, the antibacterial activity of carvacrol depends on its concentration and exposure time. And carvacrol exerts its bactericidal ability by altering the permeability of the cell membrane to cations such as H+ and K+. The disruption of ion gradients leads to damage in crucial cellular processes, ultimately resulting in cell death. However, this traditional bactericidal mechanism does not eliminate its pathogenicity. Therefore, our research aims to use carvacrol as a natural QS inhibitor and apply it to attenuate the pathogenicity of A. hydrophila NJ-35 because carvacrol could reduce the toxicity of P. aeruginosa, Salmonella enterica subsp. Typhimurium DT104, and S. aureus by inhibit their QS.

Round 2
Reviewer 1 Report
The authors improved the manuscript significantly. There are some minor spelling errors present, and references don’t follow the journal instruction, check the outputs of all references, and the journal name must be abbreviated, but otherwise, the article is scientifically sound and much easier to follow.
Minor editing of the english language is required.
Author Response
Q1: The authors improved the manuscript significantly. There are some minor spelling errors present, and references don’t follow the journal instruction, check the outputs of all references, and the journal name must be abbreviated, but otherwise, the article is scientifically sound and much easier to follow.
A1: Thanks for your advice. We have improved the language of the manuscript carefully to avoid grammatical, and bibliographic errors.
Q2: Minor editing of the english language is required.
A2: Thanks for your advice. We have revised our English writing

Reviewer 2 Report
I have several recommendations and I kindly ask authors follow them.
Lines 23-24: Should be “QS inhibition” instead of “QS inhibition mechanism”.
Lines 79-80: The phrase “…to investigate the mechanism of carvacrol reducing pathogenicity of A. hydrophila by carvacrol..” should be rephrase
Lines 287-295: Я рекомендую текст, который авторы вставили, немного модифицировать. Я предлагаю такой вариант: “The bactericidal activity of carvacrol depends on its concentration and exposure time. Certain concentrations of carvacrol corresponding to MIC provoke the death of different bacterial cells. Carvacrol exerts its bactericidal ability by altering the permeability of the cell membrane to cations such as H+ and K+. The disruption of ion gradients leads to damage in crucial cellular processes, ultimately resulting in cell death [36]. The carvacrol concentrations lower than MIC can not result in death of cells but can reduce their pathogenicity by inhibiting their QS. That was shown for P. aeruginosa, Salmonella enterica subsp, Typhimurium DT104, and S. aureus [22, 37, 38]. Our research was aimed at the use of carvacrol as a natural QS inhibitor which can be applied to attenuate the pathogenicity of A. hydrophila NJ-35. The definite concentrations resulting in such effect were revealed.»
Lines 306,308,310,312,332,342, 344: Should be “signal molecules of AHL” instead of “signal molecular AHL”.
Line 337-338: Should be “the possible action of carvacrol reducing pathogenicity of A. hydrophila NJ-35 was proposed” instead of “the mechanism of carvacrol reducing pathogenicity of A. hydrophila NJ-35 was proposed”. There is no real mechanism. The supposition of action is.
Line 340: Should be “QS inhibition, decreasing the synthesis“ instead of “QS inhibition mechanism, decreasing the synthesis”.
Figure 8. Should be “The possible action of carvacrol alleviate the pathogenicity of A. hydrophila NJ-35.” instead of “The possible mechanism of carvacrol alleviate the pathogenicity of A. hydrophila NJ-35.”
Line 352-353: Should be “QS inhibition for decreasing the pathogenicity” instead of “QS inhibition mechanism for decreasing the pathogenicity”.
Line 356: Should be “using QS inhibition to treat fish diseases.” Instead of “using QS inhibition mechanism to treat fish diseases”.
Common remark: all numbers of references should not be given as upper index in the text.
Minor editing of English language required
Author Response
Q1: Lines 23-24: Should be “QS inhibition” instead of “QS inhibition mechanism”.
A1: Thanks for your advice. We have replaced “QS inhibition mechanism” with QS inhibition” in the text.
Q2: Lines 79-80: The phrase “…to investigate the mechanism of CARVACROL reducing pathogenicity of A. hydrophila by CARVACROL..” should be rephrase
A2: Thanks for your advice. We have rephased the setence as fowllows.
Moreover, to investigate the mechanism of carvacrol reducing pathogenicity of A. hydrophila, we studied the concentration changes of AHL signal molecules in A. hydrophila QS system, as well as the expression of related genes.
Q3: Lines 287-295: Я рекомендую текст, который авторы вставили, немного модифицировать.Я предлагаю такой вариант: “The bactericidal activity of carvacrol depends on its concentration and exposure time. Certain concentrations of carvacrol corresponding to MIC provoke the death of different bacterial cells. Carvacrol exerts its bactericidal ability by altering the permeability of the cell membrane to cations such as H+ and K+. The disruption of ion gradients leads to damage in crucial cellular processes, ultimately resulting in cell death [36]. The carvacrol concentrations lower than MIC can not result in death of cells but can reduce their pathogenicity by inhibiting their QS. That was shown for P. aeruginosa, Salmonella enterica subsp, Typhimurium DT104, and S. Aureus [22, 37, 38]. Our research was aimed at the use of carvacrol as a natural QS inhibitor which can be applied to attenuate the pathogenicity of A. hydrophila NJ-35. The definite concentrations resulting in such effect were revealed.»
A3: Thanks for your suggestions to improve the reliability and quality of the article. We have revised the sentence in the text as follows.
The bactericidal activity of carvacrol depends on its concentration and exposure time. Certain concentrations of carvacrol corresponding to MIC provoke the death of different bacterial cells. Carvacrol exerts its bactericidal ability by altering the permeability of the cell membrane to cations such as H+ and K+. The disruption of ion gradients leads to damage in crucial cellular processes, ultimately resulting in cell death [36]. The carvacrol concentrations lower than MIC can not result in death of cells but can reduce their pathogenicity by inhibiting their QS. That was shown for P. aeruginosa, Salmonella enterica subsp, Typhimurium DT104, and S. aureus [22, 37, 38]. Our research was aimed at the use of carvacrol as a natural QS inhibitor which can be applied to attenuate the pathogenicity of A. hydrophila NJ-35. The definite concentrations resulting in such effect were revealed.
Q4: Lines 306,308,310,312,332,342, 344: Should be “signal molecules of AHL” instead of “signal molecular AHL”.
A4: Thanks for your advice. We have replaced “signal molecular AHL” with “signal molecules of AHL” in the text.
Q5: Line 337-338: Should be “the possible action of carvacrol reducing pathogenicity of A. hydrophila NJ-35 was proposed” instead of “the mechanism of carvacrol reducing pathogenicity of A. hydrophila NJ-35 was proposed”. There is no real mechanism. The supposition of action is.
A5: Thanks for your advice. According to the reviewer’s suggestion, we have replaced “the mechanism of carvacrol reducing pathogenicity of A. hydrophila NJ-35 was proposed” with “the possible action of carvacrol reducing pathogenicity of A. hydrophila NJ-35 was proposed.
Q6: Line 340: Should be “QS inhibition, decreasing the synthesis“ instead of “QS inhibition mechanism, decreasing the synthesis”.
A6: Thanks for your advice. We have replaced “QS inhibition mechanism, decreasing the synthesis” with “QS inhibition, decreasing the synthesis” in the text.
Q7: Figure 8. Should be “The possible action of carvacrol alleviate the pathogenicity of A. hydrophila NJ-35.” instead of “The possible mechanism of carvacrol alleviate the pathogenicity of A. hydrophila NJ-35.”
A7: Thanks for your advice. According to the reviewer’s suggestion, we have replaced “the mechanism of carvacrol reducing pathogenicity of A. hydrophila NJ-35 was proposed” with “the possible action of carvacrol reducing pathogenicity of A. hydrophila NJ-35 was proposed.
Q8: Line 352-353: Should be “QS inhibition for decreasing the pathogenicity” instead of “QS inhibition mechanism for decreasing the pathogenicity”.
A8: Thanks for your advice. We have replaced “QS inhibition mechanism for decreasing the pathogenicity” with “QS inhibition for decreasing the pathogenicity” in the text.
Q9: Should be “using QS inhibition to treat fish diseases.” Instead of “using QS inhibition mechanism to treat fish diseases”
A9: Thanks for the reviewer’s suggestion on the improvement of the article. We have replaced “QS inhibition mechanism to treat fish diseases” with “QS inhibition to treat fish diseases” in the text.
Q10: Common remark: all numbers of references should not be given as upper index in the text.
A10: Thanks for your advice. We have revised all numbers of references in the text.
Q11: Minor editing of English language required
A11: Thanks for your advice. We have revised our English writing
